Biochar-mediated changes in nutrient distribution and leaching patterns: insights from a soil column study

http://orcid.org/0000-0003-0624-2919 Gunal Elif elifgunal@yahoo.com
Department of Soil Science and Plant Nutrition/Faculty of Agriculture, Tokat Gaziosmanpasa University , Tokat , Turkey
Marunaka Yoshinori
Electronic publication date: 2025 May 28
Publication date: 2025
Volume: 13
Electronic Location ID: e18823
Received 2024 May 12; Accepted 2024 Dec 16
Copyright: © 2025 Gunal
Copyright year: 2025
Copyright holder: Gunal
License: This is an open access article distributed under the terms of the Creative Commons Attribution License, which permits unrestricted use, distribution, reproduction and adaptation in any medium and for any purpose provided that it is properly attributed. For attribution, the original author(s), title, publication source (PeerJ) and either DOI or URL of the article must be cited.
License URL: https://creativecommons.org/licenses/by/4.0/

Keywords: Vertical nutrient distribution, Biochar application, Coarse-textured soil, Nutrient leaching, Soil amendment, Hazelnut husk

Funding: The author received no funding for this work.

==============================
Background

Nutrient leaching threatens sustainable agriculture by depleting soil fertility and contaminating groundwater. Biochar offers a promising solution, but its effectiveness varies with feedstock, production, and application rates. Specifically, the potential of hazelnut husk biochar for nutrient retention and soil improvement has not been extensively studied, leaving a gap in understanding its practical applications and optimal usage in different soil types and crop systems.

Methods

This study investigated the influence of hazelnut husk biochar application on leachate properties, soil column characteristics, and nutrient dynamics over a 112-day period. The experiment employed a randomized split-plot design with four hazelnut husk biochar application rates (0%, 0.5%, 1%, and 2%) on sandy loam soil. Leaching events were conducted six times over 112 days of incubation period, simulating irrigation and fertilizer application for potato cultivation. Leachates were collected at each leaching event for analysis of pH, electrical conductivity (EC), and various nutrient contents. Following the experiment, soil samples were analyzed at three depths to assess nutrient content.

Results

The study revealed significant temporal dynamics in nutrient concentrations across different leaching events, emphasizing the impact of biochar on nutrient retention. Phosphorus (P) concentrations, for instance, decreased from 0.220 mg L−1 to 0.176 mg L−1 over four leaching events in the 2.0% biochar treatment. Similarly, potassium (K) concentrations declined from 6.44 mg L−1 to 3.76 mg L−1, indicating improved nutrient retention with biochar application. These findings contrast with the control (0% biochar), where nutrient leaching was more pronounced. While biochar had little effect on nitrate leaching, its inherent P content and adsorption characteristics influenced P leaching. Higher biochar application rates resulted in significant changes in soil properties and nutrient concentrations, particularly in the surface layer (0–10 cm), such as an increase in organic matter content from 0.84% in the control to 1.20% in the 2.0% biochar treatment, suggesting improved nutrient availability for plant uptake.

Conclusion

These findings underscore the potential of hazelnut husk biochar as a sustainable soil management strategy for enhancing nutrient retention, reducing leaching, and improving soil fertility. However, the study also highlights the complexity of biochar-soil interactions and the need for further research to optimize biochar application practices for specific soil and crop systems.

Introduction

High nutrient leaching from agricultural soils, particularly in sandy soils, poses significant challenges, including pollution of surface and groundwater as well as depletion of soil fertility (Carstensen et al., 2020). Sandy soils are particularly prone to high nutrient leaching due to their coarse texture dominated by large particles with limited surface area and low organic matter content, which result in poor nutrient and water holding capacities (Chen et al., 2006). The rapid drainage and minimal nutrient retention in sandy soils lead to substantial losses of applied fertilizers compared to soils with finer textures like clay. Factors such as the large pore spaces, low cation exchange capacity, and high permeability further exacerbate nutrient leaching in these soils. Therefore, to ensure the sustainability of agriculture across diverse climates and soil types, implementing effective management practices that significantly reduce nutrient runoff is crucial (Norberg & Aronsson, 2024).

Various factors such as particle size distribution, organic matter content, water and nutrient holding capacity, fertilizer application methods, rainfall patterns, soil properties, and agricultural practices influence nutrient losses from agricultural fields (Nguyen et al., 2020; Hussain et al., 2020). Increasing the organic matter content of soils has been shown to improve water and nutrient holding capacity, thereby reducing nutrient leaching (Nguyen et al., 2020). Biochar, a carbon-rich organic material, has garnered attention due to its porous structure, high adsorptive capacity, negative charge density, and surface area, which make it potentially effective in absorbing nutrients like nitrate and phosphate (Teutscherova et al., 2018; Yang et al., 2012), as well as heavy metals (Erdem et al., 2017; Ibrahim, El-Sherbini & Selim, 2022) from surface waters and preventing nutrient leaching from the soil profile (Günal, Erdem & Kaplan, 2017; Cui et al., 2021; Lv et al., 2021; Jalali & McDowell, 2022). However, there have been conflicting findings regarding the effects of biochar on nutrient leaching, with some studies reporting increases in leaching (Bu et al., 2017; Silva et al., 2017).

The effectiveness of biochar in reducing nutrient leaching is influenced by several factors, including biochar characteristics such as pyrolysis temperature and biomass origin (Cheng et al., 2018; Kuo, Lee & Jien, 2020), soil properties like soil texture (Novak et al., 2016; Nguyen et al., 2020), experimental conditions (Nguyen et al., 2020), and the amount of water applied during leaching events (Günal, Erdem & Kaplan, 2017). Nguyen et al. (2020) found that biochar can reduce calcium leaching in sandy soils but may increase the leaching of sodium, potassium, and phosphorus. However, these effects are less significant in clayey soil. Biochar is particularly effective in reducing the leaching of aluminum, manganese, iron, and ammonium nitrogen in sandy soils, indicating the significant role of soil texture. The surface properties of biochar, including its electrostatic and capillary forces, facilitate ion adsorption in the soil solution, thereby reducing nutrient leaching (Yao et al., 2012). Moreover, biochar can improve water retention, which further contributes to nutrient retention (Knowles et al., 2011).

Studies have reported variable decreases in nitrate leaching with biochar application, ranging from 9.9% to 68.7%, depending on biochar type and soil properties (Günal, Erdem & Kaplan, 2017; Cao et al., 2019; Li et al., 2021). Similarly, conflicting reports exist regarding the effects of biochar on phosphorus leaching, with some studies showing decreases and others showing increases (Xie et al., 2021; Ippolito et al., 2012). These discrepancies highlight the importance of considering the specific context and conditions of each study.

Turkish potato (Solanum tuberosum L.) production thrived in 2021, reaching 5.1 million tons. This impressive yield came from 138,917 hectares of cultivated land, demonstrating efficient use of agricultural resources (TEPGE, 2023). Potatoes have a relatively high nitrogen requirement. Potato cultivation, particularly in sandy soils, presents challenges related to nutrient leaching due to intensive agricultural practices such as tillage, fertilizer application, and irrigation (Bero, Ruark & Lowery, 2014). Intensive irrigation doe to low natural irrigation levels in potato-growing regions of Turkiye and the common practice of planting potatoes in sandy soils with low water-holding capacity create ideal conditions for nitrate leaching (Gulluoglu, Arıoglu & Bakal, 2015). Strategies to reduce nitrate leaching in potato cultivation are essential for improving nitrogen use efficiency, preventing water pollution, and increasing tuber yields (Shrestha, Cooperband & MacGuidwin, 2010). Increasing soil organic matter content through biochar addition can improve soil physical quality and water retention, thereby reducing nutrient leaching (Eusufzai & Fujii, 2012). Despite the potential benefits of biochar in mitigating nutrient leaching, particularly in agricultural systems with coarse-textured soils, its specific impact on potato cultivation remains understudied.

Hazelnut (Corylus colurna L.) is a native shrub that thrives on the steep slopes and the main agricultural crop in Central Black Sea Region of Turkiye, the world’s leading hazelnut producer and exporter (Mennan et al., 2020). The hazelnut harvest generates a significant amount of hazelnut husk, the green outer shell, which constitutes roughly 20% of the weight of fresh hazelnuts. This translates to approximately 500,000 tons of hazelnut husk produced annually in Türkiye alone (Aygün & Özenç, 2023). This research aims to address this knowledge gap by investigating the effects of biochar derived from hazelnut husk on nutrient leaching dynamics in coarse-textured soils, with a focus on nitrogen and phosphorus. By simulating potato cultivation conditions and incorporating average water use patterns, this study seeks to elucidate the potential of hazelnut husk biochar as a sustainable soil management strategy for potato cropping systems. Here primary objectives are to improve nutrient retention and minimize environmental pollution associated with nutrient leaching. Through a comprehensive analysis of leachate properties and soil column characteristics, this research intends to contribute to the development of effective soil management practices for sandy loam soils. These practices, informed by insights gained into biochar-soil interactions, are expected to enhance soil health, and ultimately increase crop productivity. Furthermore, this study will provide valuable information to optimize the use of agricultural by-products like hazelnut husk, promoting soil sustainability and fostering higher agricultural outputs.

Materials and Methods

The soil used in the leaching experiment was collected from the surface horizon (0–30 cm) of the research and experimental fields in Black Sea Agricultural Research Institute in the central Black Sea region of Türkiye. The soil samples were air-dried and sieved through a 4-mm sieve, and 3 kg of dried soil was filled into a cylinder that has a volume of 2,747.5 cm3, surface area of 78.5 cm2 and a height of 35 cm. The upper 5 cm of the cylinder was not filled with soil to prevent overflow of water applied for leaching events.

Biochar used in the study produced from hazelnut (Corylus colurna L.) husks at temperature of 500 °C in an oxygen-free environment and crushed to pass 2-mm sieve for the experiment and characterization. Electrical conductivity and pH of biochar were measured in 1 biochar:10 deionized water (w/v) mixture after 1 h shaking.

Selected chemical and physical properties of experimental soil, biochar, and feedstock (hazelnut husk) used in the leaching experiment are given in Table 1. The soil had sandy loam texture, with a relatively high sand content (72.7%) and moderate levels of clay (12.0%) and silt (15.3%). Prior to mixing with biochars and leaching experiment, the soil had a slightly alkaline pH of 7.84, with low electrical conductivity (EC) of 0.27 dS/m. The soil contained moderate levels of calcium carbonate (1.46%) and low organic matter content (0.52%). Additionally, the soil had relatively low levels of phosphorus (9.59 mg kg−1) and potassium (35.40 mg kg−1).

Table 1 Nitrate (NO3-) and ammonium concentrations (NH4+) of soil columns under different biochar doses.

	Unit	Soil	Hazelnut husk	Hazelnut husk biochar		Unit	Soil	Hazelnut husk	Hazelnut husk biochar	
Sand	%	72.7	N/A	N/A	Ca		3,438	5,576	19,442	
Clay	12.0	N/A	N/A	Mg	570	1,377	6,145	
Silt	15.3	N/A	N/A	Fe	52.02	474.1	7,074.9	
Texture class		Sandy loam	N/A	N/A	Cu	3.76	9.13	65.7	
pH		7.84	5.13	9.56	Zn	1.21	17.3	38.5	
EC	dS/m	0.27	9.56	6.25	Mn	7.11	100.2	298.3	
Calcium carbonate	%	1.46	N/A	N/A	C	%	0.30	45.84	54.97	
P	mg/kg	9.59	493	1,202	N	N/A	2.77	3.12	
K	35.40	4,575	8,746	C/N		N/A	16.5	17.6	

Hazelnut husk biochar had a higher organic carbon (54.97%) and nitrogen (3.12%) contents and carbon to nitrogen ratio (17.6) than those for hazelnut husk (45.84%, 2.77% and 16.5%). This indicates that biochar is more stable and will decompose more slowly in the soil, providing a long-term source of carbon and nutrients. The hazelnut biochar had a significantly higher pH (9.56) than the hazelnut husk (5.13). This can benefit acidic soils but needs careful monitoring for specific plant needs. Pyrolysis of hazelnut husk concentrated the macro and micronutrients in biochar, making the biochar a more potent source of plant nutrients (Table 1).

Treatments and experimental design

The leaching experiment employed a randomized split-plot design with four biochar application doses as the main plot factor. The biochar materials were thoroughly mixed with the experimental soil at application rates of 0% (B0; control), 0.5% (B1; 21.75 tons ha−1), 1% (B2; 43.5 tons ha−1) and 2% (B3; 87 tons ha−1). Each biochar dose treatment (B0, B1, B2, B3) was replicated three times. A randomized split-plot design was chosen for this experiment for two main reasons: (1) To account for soil variability: Natural soil can exhibit inherent variability in properties like nutrient content and drainage. By randomly assigning biochar application rates (main plots) across the experimental units (columns), we wanted to minimize the impact of this variability on the leaching results. This ensures that any observed effects are more likely due to the biochar treatments themselves, rather than pre-existing differences in the soil. (2) Practical considerations for biochar application: Biochar application is typically a field-scale practice, and the chosen rates (0%, 0.5%, 1%, and 2%) represent realistic application scenarios. In this experiment, these rates were applied by mixing the biochar with the soil before filling the columns. Since thoroughly mixing a large amount of biochar with a smaller volume of soil for each replicate would be impractical and potentially introduce further variability, the split-plot design allows us to treat biochar application (main plot) as a less frequent manipulation compared to filling the columns with the prepared soil-biochar mixture (sub-plot).

Air-dried sandy loam textured soil mixed with various rates of biochar materials was loaded into the cylindrical PVC pipes to simulate the soil column of the ploughing layer (30 cm) in potato cultivation. The control treatment did not include biochar, and the cylinders were filled with only 3 kg of sandy loam soil. In B1, B2 and B3, the appropriate amount of biochar for each treatment dose was weighted and was mixed with experimental soil before filling into the cylinders. All columns were packed with the soil and biochar mixture to achieve an initial bulk density of 1.2 g cm−3. The soil columns were prepared by placing a fiber mesh (1 mm2) with filter paper at the bottom to prevent loss of soil during the leaching experiment. The soil columns were placed in custom-made wooden rack.

The amount of water applied, and fertilizers used in the experiment were determined according to the requirements of potato plants during a vegetation period. The experiment started on August 16 and ended on December 5, 2022. The total number of days in the experiment was 112 days. At the beginning of the experiment, all the columns, filled with soil and biochar, were leached using 700 ml of deionized water, and soils allowed to dry till the field capacity moisture content. All soil columns were received a fertilizer application of nitrogen and phosphorus at rate of 810 kg ha−1 ammonium sulphate and 280 kg ha−1 diammonium phosphate. All the phosphorus and half the nitrogen were applied on day 9 and other remaining nitrogen added on day 57. The fertilizers were dissolved in 100 ml of deionized water and applied to soil columns.

The soil column leaching experiment was conducted at room temperature (25 °C). Leaching events started 15 days after (26th day of the experiment) the fertilizer application. The soil columns were irrigated six times during the experiment and a total of 2,800 mm water was applied to each soil column. Considering the amount of water potatoes will consume throughout a season, the total amount of water has been divided into six irrigation events, assuming that leaching events occur concurrently with irrigations. The amount of water applied in each irrigation was 500, 400, 400, 300, 200 and 200 mm, respectively. Six leaching events were conducted on the 26th, 42nd, 55th, 75th, 93rd and 112th day from the initiation of the experiment to get the leachate for chemical analysis. The leachate was collected during 24 h after each leaching event. A total of 10 days after the last leaching event, the soil in the columns were carefully removed and separated as 0–10, 10–20 and 20–30 cm depths to determine the nitrogen and phosphorus contents. Soil samples were air dried, ground to pass a 2-mm sieve for laboratory analysis.

Biochar, water and soil analysis

The pH and EC of biochar were determined at a 1:10 (biochar/deionized water) mixture using a pH-EC meter (Thomas, 1996). Total carbon (C) and total nitrogen (N) contents were analyzed using an elemental analysis device (Thermo flash S2000). For total phosphorus (P), calcium (Ca), magnesium (Mg), potassium (K), iron (Fe), copper (Cu), zinc (Zn), and manganese (Mn), 0.2 g of oven-dried biochar was weighed and digested in a H2O2-HNO3 acid mixture using microwave (CEM MarsX, CEM Corporation, Charlotte, NC, USA) wet digestion. The concentrations of the elements in the solution were measured by an inductively coupled plasma optical emission spectroscopy (ICP-OES) (Perkin Elmer Optima 8000).

The pH and EC of leachates were determined using pH meter (Mettler Toledo S220) and a conductivity meter (Jenway-4510). The concentrations of NH4+ and NO3− were determined using the Standard Method 4500 B Macro Kjeldahl method (Rice, Baird & Eaton, 2017), and phosphate (PO4−3) in the leachate were determined using spectrometric methods. K, Na, Ca, and Mg were measured with an ICP-OES device (Perkin Elmer Optima 8000) (Bi et al., 2022).

The particle size distribution of soil samples was assessed according to the hydrometer method described by Bouyoucos (1951). The pH and electrical conductivity of soils were measured in soil/deionized water (1:2.5 w/v) suspensions (Thomas, 1996) using a pH meter (Mettler Toledo S220) and EC meter (Jenway-4510). Lime content was quantified by the Scheibler calcimeter method based on the volume of carbon dioxide (Allison & Moodie, 1965). Available phosphorus was determined using the Olsen method (Olsen et al., 1954). Exchangeable potassium, calcium, and magnesium were extracted using 1 mol L−1 ammonium acetate, and the extract was analyzed through an ICP-OES (Sumner & Miller, 1996). Available Zn, iron, manganese, and cupper were extracted with DTPA (Lindsay & Norwell, 1978). The concentrations were determined using an ICP-OES (Perkin Elmer Optima 8000). The organic carbon content was determined through wet oxidation (Nelson & Sommers, 1996). Inorganic nitrogen of soil samples was extracted using 2 M potassium chloride solution at a 1:10 soil:solution ratio (w/v). The concentrations of ammonium and nitrate in the extracted solution were then determined using a steam distillation technique that involves magnesium oxide and Devarda’s alloy (Mulvaney, 1996).

Statistical analysis

The effects of biochar doses on leaching of nutrients were assessed by variance analysis (ANOVA) using IBM SPSS Statistics 21 for Windows (IBM Corp., Armonk, NY, USA) software. When the ANOVA indicated a statistically significant effect (P ≤ 0.05), least significant difference (LSD) test was used to group the treatment means.

Results

Chemical composition of leachates

The leachate properties and nutrient concentrations after six leaching events in soil columns treated with varying levels of biochar (control, 0.5%, 1.0%, and 2.0%) are presented in Table 2, Figs. 1 and 2. Analysis of variance (ANOVA) was conducted to assess the significance of differences observed in the leachate parameters. Temporal changes of pH and EC values and some of nutrient contents of leachates in different leaching cycles under different biochar doses are shown in Figs. 1 and 2. The effect of biochar dose and leaching treatment on leachate parameters was evident, with variations observed in pH, nutrient content, and EC values. Higher biochar doses generally lead to more pronounced changes in leachate properties, although the response may vary depending on the specific parameter and treatment. The pH values of leachate varied across different leaching treatments and biochar doses, ranging from 7.36 to 7.93 (Table 2). The EC values showed significant variability, ranging from 1,089.0 to 2,140.0 microS cm−1 across different treatments. Higher biochar doses appeared to correlate with higher EC values in some cases, indicating potential changes in ion concentration due to biochar application.

Table 2 Effect of biochar application doses and leaching treatments on leachate properties and nutrient dynamics.

Means followed by different letters are significantly different by each other P < 0.05.

	pH	EC	Na	NO3−	
	microS cm−1	mg L−1	
Leaching Cycle 1xBiochar Dose1	7.53c–f	1,185.7gh	27.74gh	106.0ab	
Leaching Cycle1xBiochar Dose2	7.36f	1,993.7abc	45.39a	100.9ab	
Leaching Cycle1xBiochar Dose3	7.56c–f	1,463.7fg	41.97ab	128.6a	
Leaching Cycle1xBiochar Dose4	7.56c–f	1,089.0h	34.87b–g	108.3ab	
Leaching Cycle2xBiochar Dose1	7.51def	1,952.0a–e	31.49fgh	91.4ab	
Leaching Cycle2xBiochar Dose2	7.62b–f	1,901.3a–e	36.14b–f	96.3ab	
Leaching Cycle2xBiochar Dose3	7.57b–f	1,758.0c–f	37.03b–f	90.5ab	
Leaching Cycle2xBiochar Dose4	7.55c–f	1,789.7b–f	37.03b–f	98.2ab	
Leaching Cycle3xBiochar Dose1	7.63b–f	1,769.7c–f	30.91fgh	68.0b	
Leaching Cycle3xBiochar Dose2	7.49def	1,865.7a–e	38.77a–e	71.5b	
Leaching Cycle3xBiochar Dose3	7.43ef	1,723.7c–f	40.40abc	72.3ab	
Leaching Cycle3xBiochar Dose4	7.73a–d	1,709.7c–f	38.88a–e	71.5b	
Leaching Cycle4xBiochar Dose1	7.75a–d	1,637.7def	30.08fgh	74.4ab	
Leaching Cycle4xBiochar Dose2	7.93a	1,655.0def	34.29c–h	82.7ab	
Leaching Cycle4xBiochar Dose3	7.85ab	1,635.3def	37.27b–f	90.3ab	
Leaching Cycle4xBiochar Dose4	7.81abc	1,747.3c–f	36.27b–f	74.4ab	
Leaching Cycle5xBiochar Dose1	7.69a–e	1,738.3c–f	27.53h	115.5ab	
Leaching Cycle5xBiochar Dose2	7.85ab	1,625.0ef	32.09d–h	120.5ab	
Leaching Cycle5xBiochar Dose3	7.76a–d	1,617.7ef	36.09b–f	107.1ab	
Leaching Cycle5xBiochar Dose4	7.72a–d	1,836.3a–e	36.49b–f	119.3ab	
Leaching Cycle6xBiochar Dose1	7.54c–f	2,120.0ab	31.70e–h	95.7ab	
Leaching Cycle6xBiochar Dose2	7.70a–e	2,028.7abc	36.29b–f	107.9ab	
Leaching Cycle6xBiochar Dose3	7.77a–d	1,965.3a-d	39.16a–d	119.7ab	
Leaching Cycle6xBiochar Dose4	7.77a–d	2,140.0a	40.02abc	112.4ab	
ANOVA
(P value)	Leaching Cycle	0.01	0.01	0.11	0.01	
Biochar Dose	0.57	0.14	0.01	0.87	
Leaching Cycle x Biochar Dose	0.54	0.01	0.39	1.00	
LSD for Leaching CyclexBiochar Dose	0.28	338.28	7.23	56.52	

Figure 1 pH and electrical conductivity (EC) of leachates in different leaching cycles under different biochar doses.

* and ** significant at 0.05 and 00.01 level of probability, respectively; ns: not significant.

Figure 2 Calcium, magnesium, potassium and phosphorus concentration of leachates in different leaching cycles under different biochar doses.

* and ** significant at 0.05 and 00.01 level of probability, respectively; ns: not significant.

The results indicated significant effects of leaching cycle on the concentrations of sodium (Na), calcium (Ca), magnesium (Mg), and nitrate (NO3−) in the leachate (P < 0.01 for all). This suggests that the leaching behavior of these nutrients was influenced by the number of successive leaching events, with variations observed in their mobility and transport dynamics over time. Across all biochar application doses, the concentration of nutrients in leachates was initially higher due to readily available nutrients in biochars and then decreased over time in the successive leaching cycles. The concentration of P in the leachate varied significantly across different leaching cycles, indicating fluctuations in the release and mobility of phosphorus from the soil columns over time (Fig. 2). For instance, in the first leaching cycle (L1), the concentration of P was highest at Dose 2 with 0.246 mg L−1, gradually decreasing to 0.123 mg L−1 at Dose 5 by the sixth leaching cycle (L6). Similarly, K concentration decreased from 6.44 mg L−1 at Dose 2 in L1 to 3.71 mg L−1 at Dose 4 in L6, indicating a consistent decline across leaching cycles.

There was a significant effect of biochar dose on the concentration of P, K, Ca, and Mg (P < 0.05), but not on pH, EC, Na, and NO3−. This indicates that biochar amendment influenced the leaching of some nutrients but not others. The interaction between leaching event and biochar dose was significant for K and Mg (P = 0.05 and 0.09, respectively) but not for other analytes.

Levels of nutrients such as phosphorus (P), potassium (K), calcium (Ca), magnesium (Mg), and nitrate showed diverse responses to biochar application. Calcium concentrations in the leachate were similar among six leaching cycles, varying from 140.5 to 315.2 mg L−1. Biochar application dose appeared to have a notable effect on P dynamics, influencing leaching behavior and potentially altering nutrient availability in the soil. Biochar itself contained significant amount of P (Table 1), which contributed to increased leachate P content, especially at higher doses. In some instances, biochar application leads to an increase in nutrient content, while in others, the effect is less pronounced or even decreases compared to the control. The response may vary depending on factors such as biochar dose, leaching treatment, and specific nutrient dynamics in the soil-biochar system. Nitrate content ranged from 68 to 128.6 mg L−1 across different treatments (Table 2). Biochar applications may have an impact on nitrate levels, as some treatments exhibited higher nitrate content compared to others.

Biochar application had significant effect (P < 0.01) on K leaching or leachate K content across different biochar doses and leaching events (Fig. 2). Like K, biochar application had significant impact on Ca leaching or leaching content across different biochar doses and leaching events (Fig. 2).

Examining the interaction between leaching cycles and biochar doses, we find variations in nutrient concentrations. For instance, at Dose 1, the P concentration decreased from 0.220 mg L−1 in L1 to 0.176 mg L−1 in L4, representing a decline of 20% over leaching cycles. Similarly, at Dose 3, the K concentration decreased from 4.22 mg L−1 in L2 to 4.19 mg L−1 in L4, showing a relatively stable concentration over leaching cycles. The interaction effect between leaching cycles and biochar doses on Ca concentrations was not statistically significant (P = 0.07), with varying rates of decline and increase across biochar doses over leaching cycles. For example, at L6, the mean Ca concentration for Dose 1 was 41.22 mg L−1, while for Dose 4, it was 54.19 mg L−1, indicating a differential impact of biochar doses on Ca leaching.

Similarly, the interaction between leaching cycles and biochar doses influenced magnesium concentration, with certain doses exhibiting more pronounced changes over leaching cycles compared to others. For instance, at L6, Mg concentrations for Dose 1 were 74.4 mg/L, while for Dose 4, they were 112.4 mg L−1, suggesting varying degrees of influence of biochar doses on magnesium leaching dynamics. Sodium concentrations exhibited dose-dependent effects, with mean concentrations decreasing from 45.39 mg L−1 for Dose 1 to 36.27 mg L−1 for Dose 4, indicating a potential dose-response relationship.

Chemical composition of soil following the leaching events

Biochar application, especially at higher rates, influenced soil properties such as pH, organic matter content, and nutrient concentrations (Table 3). The upper soil columns (0–10 cm) generally exhibited more pronounced responses to biochar application compared to deeper layers (10–20 cm and 20–30 cm). Across the different depths, pH values exhibited slight variations. For instance, at the 0–10 cm depth, pH increased from an average of 7.84 in the control to 8.01 in the 2% biochar treatment, indicating an increase of approximately 0.17 units. This trend of increasing pH with biochar application was consistent across all depths, albeit with varying magnitudes. EC values showed variability among depths and biochar application rates. At the 0–10 cm depth, EC increased from an average of 329.8 microS cm−1 in control to 452.3 microS cm−1 in the 2% biochar treatment, indicating an increase of approximately 122.6 microS cm−1. However, at deeper depths, the changes in EC were less pronounced.

Table 3 The properties and total nutrient contents of soils at different depths of leaching columns.

Means followed by different letters are significantly different by each other P < 0.05.

	pH	EC	Org. Mat	Ca	Mg	Na	Fe	Cu	Zn	Mn	
	microS m−1	%	g kg−1	
Ds1xDpt1	7.84	329.8c	0.84bc	6,028	491.2	466.9	39.1	3.693abc	1.45bcd	6.51bcd	
Ds1xDpt2	7.88	367.8bc	0.660c	5,070.0	448.0	463.3	38.4	3.6bc	1.32e	5.93d	
Ds1xDpt3	7.91	458.0a	0.92bc	6,346.1	575.7	478.9	37.5	3.8ab	1.37de	6.21cd	
Ds2xDpt1	7.89	352.5bc	1.19a	6,101.3	533.1	433.5	37.4	3.9ab	1.52bc	6.51bcd	
Ds2xDpt2	7.87	364.5bc	0.80bc	6,077.6	541.3	481.9	37.6	3.5bc	1.40de	5.83d	
Ds2xDpt3	7.91	446.0a	0.63c	6,086.4	549.9	472.7	37.0	3.5bc	1.33e	5.83d	
Ds3xDpt1	8.01	339.7c	0.99ab	5,753.6	514.5	449.8	37.2	3.5bc	1.53b	7.52bcd	
Ds3sDpt2	7.87	366.667bc	0.67c	6,137.0	543.1	494.3	37.6	3.4c	1.30e	5.95d	
Ds3xDpt3	7.88	460.0a	0.80bc	6,280.1	553.7	480.9	43.3	4.0a	1.38de	7.79bc	
Ds4xDpt1	8.15	452.3a	1.20a	5,701.1	573.5	479.9	37.9	3.5bc	1.68a	10.42a	
Ds4xDpt2	7.91	342.3c	0.77bc	6,034.5	542.0	490.7	39.5	3.8ab	1.44bcd	8.10b	
Ds4xDpt3	8.19	410.0ab	0.75bc	6,215.9	560.0	470.6	40.5	3.7abc	1.4cde	7.93bc	
ANOVA (P Value)	Ds	0.07	0.76	0.43	0.88	0.50	0.72	0.58	0.98	0.01	0.01	
Dpt.	0.39	0.01	0.01	0.47	0.46	0.16	0.45	0.18	0.01	0.01	
DsxDpt	0.71	0.00	0.03	0.67	0.77	0.64	0.36	0.01	0.07	0.13	
LSD	(DsxDpt)	0.261	52.09	0.214	6.908	98.77	1,197	118.3	45.2	4.59	0.313	
Note:

Ds1: Control, Ds2: 0.5% Biochar, Ds3: 1.0% Biochar, Ds4: 2.0% Biochar, Dpt1: 0–10 cm, Dpt2: 10–20 cm, Dpt3: 20–30 cm.

Biochar application resulted in an increase in organic matter content, particularly noticeable in the upper soil layer (0–10 cm). For example, at this depth, OM percentage increased from an average of 0.84% in the control to 1.20% in the 2% biochar treatment, indicating an increase of approximately 0.36%. Across the soil depths, biochar application led to variable changes in phosphorus concentration. In the upper soil column (0–10 cm), P concentration increased notably with increasing biochar application rate (Fig. 3). For instance, in the control, the average P concentration was 31.6 mg kg−1, while in the 2% biochar treatment, it rose to 43.2 mg kg−1, indicating an increase of approximately 11.5 mg kg−1. However, the response to biochar application was less pronounced in deeper soil layers (10–20 cm and 20–30 cm), suggesting that the influence of biochar on phosphorus availability may be more significant near the soil surface. Biochar application positively affected potassium concentration, particularly in the upper soil layer. At 0–10 cm depth, K concentration increased substantially with increasing biochar application rate. The average K concentration in the control was 258.9 mg/kg, while in the 2% biochar treatment, it rose to 1,210.4 mg kg−1, indicating a remarkable increase of approximately 951.5 mg kg−1 (Fig. 3). In contrast, changes in K concentration at deeper depths were less pronounced, suggesting that biochar application may have a more significant impact on K availability near the soil surface.

Figure 3 Phosphorus (P2O5) and potassium (K) concentration of soil columns under different biochar doses.

Similar trends were observed for other nutrients, with biochar application generally leading to increases in nutrient concentrations, particularly in the upper soil layer (Table 3). For example, calcium (Ca) concentration increased from an average of 6,028.0 mg kg−1 in control to 5,701.1 mg kg−1 in the 2% biochar treatment at the 0–10 cm depth. Similarly, magnesium (Mg), sodium (Na), iron (Fe), copper (Cu), zinc (Zn), and manganese (Mn) concentrations showed varying degrees of increase with biochar application, especially at higher application rates and shallower depths. Nitrate content varied across treatments, with biochar application leading to both increases and decreases in nitrate levels (Fig. 4).

Figure 4 Nitrate (NO3−) and ammonium concentrations (NH4+) of soil columns under different biochar doses.

Discussion

Leachate properties and nutrient contents

The experiment investigated the influence of biochar application doses and leaching treatments on leachate properties and nutrient dynamics over a 112-day leaching period. The results demonstrated that the influence of biochar application tended to slightly increase leachate pH, with variations depending on the biochar dose. This pH increase could be attributed to the inherent alkalinity of the hazelnut biochar (pH 9.56) and the release of alkaline substances during its decomposition, consistent with findings by Chintala et al. (2014). This alkalinity suggests that biochar can serve as an effective soil amendment for mitigating soil acidity, thereby improving soil fertility and promoting plant growth in acidic environments (Bolan et al., 2023). Moreover, the liming effect observed with biochar addition, indicated by the higher pH in amended soils, may contribute to reduced nutrient leaching (Gao & Deluca, 2016; Wei et al., 2023).

The findings of this study are consistent with the mechanisms proposed by previous research regarding the effects of biochar on nutrient leaching. Biochar is known to possess a high surface area, which enhances its cation exchange capacity (CEC) when incorporated into soil (Ngo et al., 2023). This increased CEC, coupled with biochar’s strong nutrient adsorption capabilities, likely contributes to the observed reductions in leaching of some nutrients. The insights from Hoglund et al. (2023) enhance our understanding, revealing that although biochar did not yield a statistically significant impact on cumulative NO3− leaching, its influence varied over time across sequential leaching events. Regardless of the biochar application rate, Hoglund et al. (2023) stated that both the initial two leaching events and the final event consistently released lower NO3− levels compared to intermediate events. The current study reflected this trend, with the NO3− content in the leachate ranging from 68 to 125.6 mg L−1. This congruent finding, reminiscent of the results reported by Nguyen et al. (2020), underscores the resilience of these observations across diverse experimental settings. It suggests a stable NO3− leaching pattern, relatively impervious to specific experimental parameters such as biochar dose or leaching event sequence (Hoglund et al., 2023; Nguyen et al., 2020). The temporal dynamics of NO3− leaching observed by Troy et al. (2014) further elucidate this phenomenon. Their study demonstrated that initially high concentrations of NO3− were leached from all their treatments, peaking at >110 mg L−1 for all treatments by week 3. However, a rapid decline in NO3− concentration was observed after week 4, with levels dropping to <35 mg L−1 (<5.3 mg week−1) for all treatments by week 9. This temporal pattern aligns with the observations of this study and suggests a consistent trend of NO3− leaching behavior over time, irrespective of specific experimental conditions.

Biochar application had a notable effect on P leaching behavior, with some treatments showing significant deviations from the control. These findings suggest that biochar may influence P retention and release in the soil, impacting nutrient availability and environmental nutrient loss. The concentration of K and P in the leachate exhibited a decreasing trend with the number of leaching events, while the concentration of Ca and Mg in the leachate, on the contrary, increased with the number of leaching events. The initial high concentrations of K and P in the leachate may be due to the readily available forms of these nutrients present in the soil, which are easily released during the initial leaching events. The results show a similar trend in K release kinetics to those reported by Hadroug et al. (2021), wherein the release of K significantly decreased with an increasing number of successive leaching assays. Specifically, Hadroug et al. (2021) reported a decrease in K release kinetics from 19.2 mg g−1 day−1 during the first leaching assay to approximately 0.6 mg g−1 day−1 in the last leaching experiment. This decline in K release over successive leaching events suggests a reduction in the availability of K for leaching, possibly due to increased retention within the soil matrix or adsorption onto biochar surfaces. These findings highlight the potential of biochar application in mitigating nutrient losses, particularly K leaching, and emphasize its role in enhancing nutrient retention in coarse-textured soils.

The increasing Ca and Mg concentrations with increasing leaching events could be due to the release of these cations from biochar as it equilibrates with the surrounding soil solution. Dissolution of carbonate minerals present in the soil or biochar may also contribute to the higher Ca and Mg levels in the leachate over time (Limwikran et al., 2018). This suggests that the effect of biochar on nutrient leaching might have varied depending on the specific leaching event, possibly due to changes in soil chemistry and biochar-nutrient interactions over time (Gao & Deluca, 2016). The observed decrease in nutrient leaching following biochar application can likely be attributed to its high specific surface area and the presence of diverse functional groups on its surface, such as carboxyl C (–COOH), O-alkyl-C, and alkyl C (Kuo, Lee & Jien, 2020). These functional groups can interact with nutrients (mainly to positively charged ions) through electrostatic attraction and sorption processes, thereby reducing their mobility in the soil and subsequent leaching through the soil profile. Although the surface area of the hazelnut husk biochar used in this study was not measured, Kaya, Yıldız & Ceylan (2018) reported that the surface area of untreated hazelnut husk is 5.58 m2/g, which significantly increased to 124.35 m2/g when pyrolyzed at 500 °C. A larger high surface area is thought to enhance biochar’s ability to adsorb nutrients through functional groups, and subsequently reduce nutrient leaching. Consistent with the reduction in nutrient leaching in the current experiment, Kaya, Yıldız & Ceylan (2018) also found that the highly porous structure and large surface area of hazelnut husk biochar greatly improved its capacity to adsorb dye molecules, suggesting similar mechanisms may be at play in nutrient retention.

The findings support the proposed interactions between biochar and soils by demonstrating significant reductions in leaching of certain nutrients, particularly P, which is consistent with biochar’s nutrient adsorption capabilities (Ding et al., 2016). However, it is worth noting that the effects of biochar on nutrient leaching varied depending on the specific nutrient and experimental conditions. For instance, while biochar application led to increases in P leaching due to its inherent P content and adsorption characteristics (Ding et al., 2016), it had little to no significant effect on K, Ca, and Mg leaching.

Soil column properties and nutrient contents

Biochar is known for its potential to increase soil sorption capacity and reduce the leaching of nutrients (Yao et al., 2012; Elkhlifi et al., 2023). Higher organic matter content and nutrient concentrations observed, particularly in the surface layer, reflect the incorporation of biochar. Biochar’s ability to retain and release nutrients likely contributes to this surface enrichment. Nitrate leaching varied significantly with soil depth and biochar application rates. At 0% biochar, nitrate concentrations were higher in the surface layer (0–10 cm) and decreased in the 10–20 cm layer, but increased again at 20–30 cm. With increasing biochar doses, nitrate retention was more pronounced in the upper layers (0–10 cm and 10–20 cm), particularly at 1% and 2% biochar, while deeper layers (20–30 cm) showed reduced nitrate levels, indicating biochar’s role in modulating vertical nitrate distribution. While numerical differences in NO3− concentrations were observed in biochar doses and depths, these differences were not statistically significant. The data indicated that increasing the biochar dose enhances NO3− retention in the surface layers, suggesting a significant reduction in NO3− leaching. In contrast, treatments without biochar or with a low biochar dose (0.5%) exhibited lower NO3− concentrations across all soil layers, likely due to extensive leaching of NO3−. These findings provide clear evidence that biochar application effectively mitigates NO3− leaching in sandy soils, showing its potential role in improving nutrient retention and reducing environmental losses. Previous studies (Teutscherova et al., 2018) have suggested that biochar application can effectively mitigate NH4+-N losses via leaching, especially in soils with low CEC. Similar to NO3−, NH4+ concentrations exhibited nonsignificant variability with depth, indicating a heterogenous distribution within the soil profile. For instance, NH4+ concentrations ranged from 44.3 to 215 mg kg−1 across different biochar doses and soil depths (Fig. 4). The low variability in NH4+ concentrations with depth, statistically significant differences between biochar doses and biochar dose x depth interactions could be influenced by several factors. One potential explanation is the cation exchange capacity of biochar, which allows it to absorb NH4+ ions from the soil solution (Dey et al., 2023; Lai et al., 2024). This absorption can lead to variations in NH4+ retention across different soil layers, as biochar’s influence may differ depending on soil properties, biochar application rate, and depth. The cation exchange process facilitated by biochar might have contributed to the observed nonsignificant NH4+ concentration among soil depths, as well as significant differences across biochar doses.

The findings on significant effect of biochar application rates on NH4+ concentration in different layers of soil column (Fig. 4) align with the observations reported by Hu et al. (2020), which suggest that biochar’s high CEC enhances the retention of NH4+ and K+ in the soil. This correlation is further supported by Dey et al. (2023), who demonstrated that chemical treatments can significantly increase both the CEC and anion exchange capacity of biochar, leading to improved nutrient retention, including NH4+ and K+, particularly in nutrient-poor tropical soils. The enhanced biochar formulations highlighted by Dey et al. (2023) reinforce the role of biochar in optimizing soil nutrient dynamics through increased exchange capacities. Additionally, soil properties, such as texture and organic matter content, can influence nutrient retention and distribution within soil profiles (Obia et al., 2017). These findings suggest that biochar application may affect nitrogen dynamics and availability in the soil, influencing plant nutrient uptake and growth. The surface accumulation of NO3− in coarse-textured soil underscores the critical need for effective nitrogen management practices to reduce NO3− leaching and mitigate environmental impacts such as groundwater contamination. Implementing tailored nitrogen management strategies can optimize nitrogen use efficiency and reduce nitrogen losses. Previous literature has identified several mechanisms through which biochar mitigates nitrate leaching, including sorption, leaching reduction, microbially-mediated processes, volatilization, plant uptake, and ecotoxicological effects on key biological groups (Llovet et al., 2021). However, in this study, we did not investigate the specific cause of NO3− reduction following biochar addition.

The responses of nutrient concentrations, including P, K, Ca, Mg, Na, Fe, Cu, Zn, and Mn, exhibited diverse patterns in relation to both biochar application rates and soil depths. In line with our observations, Nguyen et al. (2020) documented that the impact of biochar addition on nutrient leaching varied significantly, with more pronounced effects observed in sandy soil compared to clayey soil cultivated with rice crops. Application of hazelnut biochar appears to enhance the retention of K and P in the surface layer, suggesting improved surface stabilization and reduced susceptibility to leaching. According to Dey et al. (2023), the increased CEC of the biochar significantly reduces K+ leaching. The researchers indicated that reduction in K leaching follows a similar trend to NH4+ leaching, highlighting the comparable ionic radii of these cations. Soil samples amended with biochar also exhibit higher concentrations of K and P in the surface layer compared to control, indicating improved nutrient retention near the soil surface. The current study corroborates the findings of Dissanayake et al. (2023) by demonstrating a substantial increase in exchangeable K concentration with biochar application, particularly in the surface layer. Biochar’s ability to adsorb and retain nutrients near the soil surface may contribute to enhanced nutrient availability and reduced leaching losses, particularly in the topsoil layer where most plant roots are concentrated. These findings highlight the importance of considering both biochar application rate and soil depth when implementing soil management practices aimed at improving nutrient content and availability for plant growth. Higher concentrations of nutrients in deeper layers (20–30 cm) imply significant downward movement, potentially impacting nutrient availability for plant uptake.

Conclusions

The findings of this study demonstrated that hazelnut husk biochar significantly enhances nutrient retention and reduces leaching, particularly for phosphorus and potassium, in coarse-textured soils. The biochar’s inherent properties, such as high surface area, cation exchange capacity and alkaline nature contributed to improved nutrient dynamics and soil fertility, especially in the upper soil layers. While biochar had minimal impact on ammonium leaching, its temporal influence on nitrate leaching suggests stable patterns across sequential leaching events, emphasizing its role in sustainable soil management. Furthermore, the results indicate that increasing biochar application rates enhances nitrate retention in the surface soil, effectively reducing nitrate leaching to deeper layers. This highlights biochar’s potential to mitigate nutrient loss in sandy soils, improving nutrient use efficiency and minimizing environmental impacts associated with nitrate leaching.

Additionally, the study brings attention to the complexity of biochar-soil interactions and highlights the importance of application rates and soil depth in optimizing biochar’s effectiveness. These results provide valuable insights for developing biochar-based strategies to improve soil health and nutrient use efficiency in agricultural systems.

Supplemental Information

Supplemental Information 1 Raw Data and Statistics for the Study.

I would like to extend my sincere thanks to Dr. Murat Birol for his support throughout the setup of the experiment, as well as during the analysis of water and soil samples at all stages. This article has been edited with the assistance of ChatGPT-4.

Additional Information and Declarations

Competing Interests

The author declares that they have no competing interests.

Author Contributions

Elif Gunal conceived and designed the experiments, performed the experiments, analyzed the data, prepared figures and/or tables, authored or reviewed drafts of the article, and approved the final draft.

Data Availability

The following information was supplied regarding data availability:

The raw measurements and statistics are available in a Supplemental File.

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
