# Peer review of "Biochar-mediated changes in nutrient distribution and leaching patterns: insights from a soil column study"

_PeerJ, doi:10.7717/peerj.18823_

## Round 0.1 · original submission · Major Revisions

If you feel you can revise your manuscript according to the reviewers' comments, please revise your manuscript and submit it. Please also send us your written responses to each of the reviewers' comments.

Yours,

Yoshi

Prof. Yoshinori Marunaka, M.D., Ph.D.

·

Basic reporting

This study focused on Biochar-mediated changes in nutrient distribution and leaching patterns: insights from a soil column study. However, the work needs some improvement before publication. Please the following comments below:

Lots of articles are already well reported in the literature, and the proposed review did not present any new directions for further research.

Experimental design

1. Add the experimental design layout
2. any specific reason authors chose the hazelnut husk for biochar

Validity of the findings

1. Effect of liming properties of biochar on soil and other parameters did not get discussed deeply
2. introduction section should be more thorough and add some latest cited articles in 2024

Reviewer 2 ·

Basic reporting

The manuscript is well-structured, and the language is generally comprehensible. However, there are several typographical and formatting errors that should be addressed. While the quality of the writing is acceptable, further improvements are necessary. The author investigates the effect of biochar on nutrient and water leaching, yet fails to characterize the physical properties of the biochar utilized in the study. Considerable attention is given to the chemical properties of biochar, which may not be directly relevant to mitigating leaching. Furthermore, the tables and figures are not adequately designed or presented.

I would recommended that the author conduct additional analyses to measure key physical parameters, such as pore size and pore density, of the biochar employed. With this data, the author can provide a more comprehensive discussion on the influence of biochar's physical characteristics on nutrient and water adsorption in sandy soil matrices. Addressing these issues would significantly enhance the quality and impact of the manuscript.

Experimental design

The experimental design is fine, but the ways to present the results should be improved.

Validity of the findings

No comment.

Reviewer 3 ·

Basic reporting

The article entitled “Biochar-mediated changes in nutrient distribution and leaching patterns: insights from a soil column study” investigated the role of nutrient leaching of biochar by applying biochar to a sandy loam soil. But, there were major flaws in this article; nutrient adsorption by biochar was an important factor in reducing nutrient leaching, but it was not studied in depth. Moreover, the improvement of soil structure by biochar is also one of the important factors, but the nutrient did not measure the indexes about soil structure. Besides that, there are many problems in the writing of the article as follows. Therefore, I suggest the authors to add some of the indicators and resubmit the article after making significant changes to the writing.

Experimental design

Mixing soil with biochar in soil columns is bound to change the original soil structure, but this study still lacks indicators of soil structure.

Validity of the findings

no comment

Additional comments

L15-16, this section should belong to the Methods, while Background should present the current status and shortcomings of this research.
L18, this section should indicate the type of biochar used, i.e. Hazelnut husk biochar.
L23-25, this section should highlight the most important results. This section can be deleted and is not the main purpose of the study.
L25-28, “Analysis of leachate chemical composition revealed significant variations in nutrient concentrations across different leaching events, with biochar application influencing phosphorus (P), potassium (K), calcium (Ca), and magnesium (Mg) dynamics.” significant variations? What's the variations, whether it is an increase or a decrease. This vague description can be removed.
L36-40, this section is missing a summary of the results, i.e. conclusion.
L49-52, this section of the introduction is not very relevant to this study and it is suggested that it be deleted and replaced with what are the causes of high nutrient leaching in sandy soils.
L62-62, the introduction states “The efficacy of biochar in reducing nutrient leaching depends on various factors, including biochar characteristics, soil properties, experimental conditions, and the type and amount of water applied during leaching events”. But the literature cited does not indicate what type of biochar was applied and what soil type it was.
L174-187, this section of the description should be moved to Materials and Methods
L190-195, this section of the description which has no practical significance should be deleted. A similar problem occurs in L243-244
There are too many sections in the discussion that duplicate the results and do not provide an in-depth discussion of possible mechanisms.
There is too much redundancy in the conclusion, which is not condensed into a highly summarized conclusion.

Reviewer 4 ·

Basic reporting

Title: Biochar-mediated changes in nutrient distribution and leaching patterns: insights from a soil column study.
The author addressed a unique topic: “Biochar-mediated changes in nutrient distribution and leaching patterns: insights from a soil column study”. The manuscript contains novel information which can further strengthen the existing knowledge of the field, particularly application of hazelnut husk biochar on leachate properties, soil column characteristics, and nutrient dynamics.
Scientists planned their study according to need of time. Moreover, the research area seems well motivated, and research findings support the claim and objectives properly made in the manuscript. Results are correctly presented and compared with existing knowledge of the field.

Experimental design

Experimental design is not clear. Why did the author employ randomized split-plot design?

Validity of the findings

The study highlights the complexity of biochar-soil interactions. I think results have some potential broader applicability.

Additional comments

Introduction is well managed but needs little improvement regarding study gap, hypothesis, and clear objectives.
Try to discuss results with recent literature and provide reasoning of the responses recorded. Improve the discussion with logical and reasoning approaches.
Follow the journals guidelines for reference style and bibliography.
Moderate editing of English language required.

---

## Round 0.2 · Minor Revisions

Please address these final minor changes and resubmit.

Yours,

Yoshi

Prof. Yoshinori Marunaka, M.D., Ph.D.

·

Basic reporting

The paper has been carefully revised according to my suggestions, , so I think it can be recommended for acceptance and publication.

Experimental design

The paper has been carefully revised according to my suggestions, , so I think it can be recommended for acceptance and publication.

Validity of the findings

The paper has been carefully revised according to my suggestions, , so I think it can be recommended for acceptance and publication.

Additional comments

The paper has been carefully revised according to my suggestions, , so I think it can be recommended for acceptance and publication.

Reviewer 3 ·

Basic reporting

The article titled “Biochar-mediated changes in nutrient distribution and leaching patterns: insights from a soil column study” has undergone major revisions and is much improved in terms of writing. However, there are still a handful of problems with the current article.
L29-42, highlights the effects of biochar application versus no biochar application on nutrient leaching. Instead, it focuses on temporal dynamics.
L492-506, the conclusion section should highlight the most significant results. E.g., delete L492-493.
The experimental design shows that there are 3 replications. However, all figures (including line and bar graphs) do not have error bars and significance analysis.

Experimental design

no comment

Validity of the findings

no comment

Reviewer 4 ·

Basic reporting

Most of the comments are addressed by the author(s).

Experimental design

Most of the comments are addressed by the author(s).

Validity of the findings

Most of the comments are addressed by the author(s).

Additional comments

It is suggested to improve the quality of figures.

---

## Round 0.3 · accepted · Accept

Congratulations!
Yours,
Yoshi
Prof. Yoshinori Marunaka, M.D., Ph.D.

Reviewer 3 ·

Basic reporting

No comment

Experimental design

No comment

Validity of the findings

No comment

Additional comments

No comment